# BAC-FISH Based Physical Map of Endangered Catfish *Clarias magur* for Chromosome Cataloguing and Gene Isolation through Positional Cloning

**DOI:** 10.3390/ijms232415958

**Published:** 2022-12-15

**Authors:** Vishwamitra Singh Baisvar, Basdeo Kushwaha, Ravindra Kumar, Murali Sanjeev Kumar, Mahender Singh, Anil Rai, Uttam Kumar Sarkar

**Affiliations:** 1ICAR—National Bureau of Fish Genetic Resources, Canal Ring Road, P.O. Dilkusha, Lucknow 226002, India; 2Division of Agricultural Bioinformatics, ICAR—Indian Agricultural Statistics Research Institute, Library Avenue, New Delhi 110012, India

**Keywords:** BAC clones, chromosome, *Clarias magur*, DNA probe, FISH

## Abstract

Construction of a physical chromosome map of a species is important for positional cloning, targeted marker development, fine mapping of genes, selection of candidate genes for molecular breeding, as well as understanding the genome organization. The genomic libraries in the form of bacterial artificial chromosome (BAC) clones are also a very useful resource for physical mapping and identification and isolation of full-length genes and the related *cis* acting elements. Some BAC-FISH based studies reported in the past were gene based physical chromosome maps of *Clarias magur* (magur) to understand the genome organization of the species and to establish the relationships with other species in respect to genes’ organization and evolution in the past. In the present study, we generated end sequences of the BAC clones and analyzed those end sequences within the scaffolds of the draft genome of magur to identify and map the genes bioinformatically for each clone. A total of 36 clones mostly possessing genes were identified and used in probe construction and their subsequent hybridization on the metaphase chromosomes of magur. This study successfully mapped all 36 specific clones on 16 chromosome pairs, out of 25 pairs of magur chromosomes. These clones are now recognized as chromosome-specific makers, which are an aid in individual chromosome identification and fine assembly of the genome sequence, and will ultimately help in developing anchored genes’ map on the chromosomes of *C. magur* for understanding their organization, inheritance of important fishery traits and evolution of magur with respect to channel catfish, zebrafish and other species.

## 1. Introduction

Genus *Clarias* (family: Clariidae) comprises 61 valid species (http://fishbase.org; accessed on 24 October 2022), and *C. magur* (commonly called magur) is an important catfish species among them with good aquaculture potential. It is found in freshwater inland wetlands of India, Bangladesh, and Nepal. It is widely distributed in the north, east, central and northeast regions of India. It is categorized as endangered (A3cde+4acde, ver. 3.1) in IUCN Red List Category and Criteria. The population of this species is severely fragmented, and the trend is decreasing. A population decline of over 50% occurred in the last few years, and the predicted decline at the same or a slightly higher rate is the reason for its endangered status. The identified threats for the species are excessive exploitation, destruction of breeding grounds due to wetland conversion, and pesticide applications in agricultural fields. Other major constraints are mass seed production of this species, unavailability of suitable feeds, disease management and lack of standardized management practices [1,2].

In the current situation, the research focus in this species is mainly on breeding and improving the growth rate and disease resistance along with management practices. Moreover, several studies have been carried out for characterization and improvement of this species. Karyological investigations have been also reported using Giemsa, C-banding, Ag-NOR, CMA_3_ staining and 18S as well as 5S rDNA-FISH [3]. Several molecular studies related to the mitochondrial genome [4,5], whole genome sequences [6], SSR markers [7], stress-responsive genes [8,9], transcriptome analysis [10], and chromosome mapping using BAC-FISH [3,11] were also undertaken in magur.

Development of a chromosome map is essential to elaborate the chromosome organization as well as the evolution of magur in comparison to related species [12]. The bacterial artificial chromosome (BAC) is an important genomic resource for maintaining large DNA inserts into bacterial plasmids, developing DNA probes and mapping on chromosomes through fluorescence in situ hybridization (FISH) for identification and further use in other genomic studies [13]. The BAC-FISH tool, using BAC as DNA probe for FISH, has been used as a marker to provide information regarding chromosome identification and rearrangements. An authentic BAC-FISH is used as a cytogenetic tool for individual chromosome identification [11], biodiversity assessment [14], marker-assisted breeding [15], genome sequencing/assembly and utilization of information in aquaculture.

The study on individual chromosome identification using BAC-FISH may take lead in establishing the foundation for genomic research to exploit more genes and genomic information for breeding and to provide powerful evidence for evolution and assembling whole genome sequences [16]. The objective of the present study was to provide the BAC-FISH based identification of magur metaphase chromosomes and to develop a well-grounded physical map of the genome of the valuable species. The study integrated the physical and anchored scaffold map construction on the genome for figuring out new chromosome rearrangements. The structural and evolutionary genomics of zebrafish and channel catfish have also been coupled with the magur for understanding chromosome evolution.

## 2. Results

### 2.1. BAC Clone Screening

The BAC library of magur genome (~1.02 Gb size) contained 55,141 clones covering ~6× genome with an average DNA insert size of 115 kb. The clones were maintained into 144 plates of 384-well format containing LB media. The metaphase chromosome complement counting revealed a total of 25 chromosome pairs, with 8 pairs of mono-armed (4 pairs subtelocentric and 4 pairs acrocentric) and 17 pairs of bi-armed (7 pairs metacentric and 10 pairs submetacentric) chromosomes.

A total of 3295 clones were sequenced, generating 6334 BAC end sequences (BESs, 3178 forward and 3156 reverse). From the end sequenced BAC clones, we selected 36 clones randomly, which were bioinformatically mapped on the magur genome scaffolds and used for BAC-FISH on magur’s metaphase chromosomes. The sizes of the forward and reverse BESs of 36 clones ranged from 159 bp (forward sequence of clone ID: 004M12) to 909 bp (reverse sequence of clone ID: 008P24), indicating good size of the generated sequences (Appendix A). The lengths of the BAC clones, estimated bioinformatically by aligning both BESs of the clones on magur genome scaffolds, ranged from 29,029 bp (ID: 003M7, present on Scaffold_1) to 1,991,239 bp (ID: 001N22, present on Scaffolds_1&1063), and both of them localized on metacentric chromosome 4. The average estimated size of the 36 BAC clones was approximately 223,616 bp. The size of scaffolds, on which both BESs of BAC clones were mapped, ranged from 445,015 bp (Scaffold_530) to 9,885,605 bp (Scaffold_1). A total of 11,082 SSRs were mined on these clones using MISA tool. All of the clones contained SSRs, and the SSR types ranged from simple to compound repeats. The BAC ID: 007F07 contained a maximum of (3548) SSRs, while ID: 003M7 was confined to a minimum (13). Analysis of 36 mapped BAC clones revealed an average GC content of 39.1% (ranging from 36.7% in clone 008N5 to 44.8% in clone 004M12). 

### 2.2. BAC Mapping on Chromosomes and Scaffolds 

The BAC-FISH signals (green and red) of 36 clones were distributed on 28 locations of 16 different chromosome pairs (Nos. 2, 4, 5, 7–18, 20) of magur metaphase spreads (Figure 1, Appendix A). Therefore, these clones may be considered as chromosome markers for identifying an individual magur chromosome. Out of 36 clones studied, 14 clones were specific to four metacentric chromosome pairs (Nos. 2, 4, 5, 7), 20 clones were specific to all 10 sm chromosome pairs (nos. 8–17) and 1 clone (ID: 010L11) was specific to st chromosome pair 20 for their identification. One clone (ID: 010P19) was not specific to a particular chromosome, as it was localized on two chromosomes, i.e., 11th sm and 18 st.

Further, the analysis of alignment information of these 36 BAC clones revealed that BESs were distributed on 33 genome scaffolds (Nos. 1, 7, 8, 19, 26, 37, 48, 49, 66, 68, 70, 75, 92, 116, 140, 143, 165, 173, 176, 206, 232, 254, 255, 268, 314, 325, 369, 530, 712, 1012, 1063, 1711, 1942) of magur (Appendix A). Eight clones (IDs: 001N22, 002J12, 002O2, 003L9, 003M7, 004F12, 007F14, 008P24) were mapped on Scaffold_1. The FISH of these clones revealed that six clones (IDs: 001N22, 002J12, 002O2, 003L9, 003M7, 004F12) were hybridized on q arm of the 4th metacentric chromosome pair, while one clone (ID: 007F14) was hybridized on the p arm of the 16th sm chromosome pair, and one clone (ID: 008P24) on the p arm of the 9th sm chromosome pair. Two clones (IDs: 004N14, 005M20) were located on Scaffold_26 and hybridized on the q arm of sm chromosome 17. Two clones (IDs: 002P12 and 005B17) were jointly located on two scaffolds (Nos. 176 and 1711) and observed to be located on the q arm of sm chromosome 8. Interestingly, BESs of seven clones (IDs: 001I2, 001N22, 002P12; 005B17, 005M20, 007A22, 007F7,) were found to be mapped on two different scaffolds (Nos. 1942&369, 1&1063, 176&1711, 1711&716, 712&26, 1012&140, 19&165, respectively), and their FISH probes were subsequently hybridized and visualized on different chromosome pairs 9 (sm), 4 (m), 8 (sm), 8 (sm), 8 (sm), 11 (sm) and 14 (sm), respectively. Thus, the BES mapping and FISH tools could be helpful in joining two or more scaffolds to obtain longer scaffolds. This way, the magur genome assembly may be improved and validated, too. One clone (ID: 010P19) was located on Scaffold_8, which was subsequently positioned on the p arms of two chromosomes (sm pair 11 and st pair 18) using FISH, indicating that this clone region may be duplicated on two different chromosomes. Three other clones (IDs: 008E4, 008F24, 008G1), localized on other scaffolds, were also mapped on the q arm of the 4th metacentric chromosome pair, while one clone (ID: 007D10, present on Scaffold_92) was localized on the p arm of the 4th chromosome pair. 

In terms of genes lying in 36 clones, 34 clones contained a total of 555 genes, ranging from one gene in two clones (IDs: 003M7, 006K13) to 210 genes in one clone (ID: 007F7), while two clones (IDs: 004M12 and 006A14) did not contain any genes. These genes contained various numbers of CDS/exons, ranging from one (in clone IDs: 003M7, 006K13) to 1605 (in clone ID: 007F7) (Appendix A). Different genes contained different CDS regions, and no relationship was observed in their numbers. The details of the genes in terms of ID, names, description, position on scaffold, strand type (+/−), number of exons and length are presented in Appendix A.

### 2.3. Gene Enrichment Analysis

In gene enrichment and relationship analysis using Panther db, a total of six pathways were hit involving 60 genes. These six pathways represented nine components involving 147 gene IDs (Figure 2). In the biological process, 14 pathways were identified, involving 60 genes with 115 process hits. The cellular process had maximum (27%) involvement, followed by the metabolic process (23%) (Figure 3a). In molecular function, six pathways were identified, where 60 genes, affecting 36 functions, were involved. Binding function had the maximum genes (15), contributing 41.7% (Figure 3b). In protein class, 14 pathways were identified, involving 43 function hits and 60 genes. Metabolic conversion enzyme involved maximum genes (9) and contribution (20.9%) (Figure 3c). In cellular component, only two pathways were identified, involving 60 genes and 46 hits, where cellular anatomical entity involved maximum genes (34) and contribution (73.9%) (Figure 3d).

### 2.4. Comparative Genomics

While aligning genes present on 36 undertaken clones of magur with zebrafish and channel catfish genomes, 31 clones containing 86 genes were in synteny in some genes with channel catfish and zebrafish, while the remaining five clones did not contain any gene having synteny with the latter two species. The details of genes along with their location on scaffolds and chromosomes in three species are presented in Appendix A. The synteny analysis of 45 genes, located on different chromosomes of channel catfish and zebrafish, was utilized and is presented in Figure 4, while another 41 genes were not considered in the synteny visualization. The 26 genes present on 10 clones (IDs: 004F12, 006O16, 007B2, 007K15, 008E4, 008F24, 008G1, 009D11, 010P13, 010P19) showed their conserved synteny with channel catfish and zebrafish, while the synteny of 60 genes distributed on 21 clones had disrupted synteny among magur, channel catfish and zebrafish with location on different sites. 

Four major circular unrooted phylogenetic trees were constructed based on *p*-distances using the NJ method for clear visualisation of common genes present on *C. magur* scaffolds and channel catfish and zebrafish chromosomes (Figure 5a–d). Phylogeny of genes present on SF_1 of *C. magur* with channel catfish and zebrafish generated six clusters (Figure 5a), while SF_26, SF_75, SF_92; SF_254, SF_325, SF_530 of *C. magur* with channel catfish and zebrafish generated four clusters (Figure 5b). Similarly, SF_48, SF_49, SF_70, SF_206, SF_232, SF_255, SF_268, SF_314 scaffolds grouped into five clusters (Figure 5c) and scaffolds SF_8, SF_140, SF_165, SF_173, SF_176, SF_369 grouped in to eight clusters with channel catfish and zebrafish (Figure 5d).

## 3. Discussion

The whole genome draft of *C. magur* (NCBI Acc. No. QNUK00000000) is a valuable genomic resource for researchers. The genes specific to environmental and terrestrial adaptation, such as urea cycle, vision, locomotion, olfactory/vomeronasal receptors, immune system, anti-microbial properties, mucus, thermoregulation, osmoregulation, air-breathing, detoxification, etc. of this catfish have been identified and analyzed [6]. The results indicated that the genome possessed several unique and duplicate genes similar to terrestrial or amphibian counterparts. 

Large-insert DNA libraries, such as BAC, are based on the F (fertility) factor of *Escherichia coli*, which has plasmid in a supercoiled circular form with low-copy number in the host cells. These libraries provide a means to clone complex genomes’ smaller DNA segments to reduce the genome complexity for generating more copies for ease of study. BAC clones are useful in genome assembly, identifying full-length genome sequences, to develop integrated physical and or genetic maps [17,18]. These are also utilized for comparative genomic studies [19,20] and are very important as an alternate source of live material for various biological applications. They were traditionally used as a DNA source for the first step of whole-genome sequencing projects [19]. Here, we constructed the first BAC library from *C. magur* genome and characterized it partially. Utilizing benefits of the genome sequence with high genome coverage in the BAC library, we screened the BAC clones of magur for further applications. This study delivers the first set of chromosome-specific cytogenetic and molecular markers in magur, which is important information for future developments in cytogenetic and genomic studies.

In a good-quality BAC library, a large proportion of the clones should contain insert-DNA and the insert size should be as large as possible, homogeneous and non-chimeric. The *C. magur* BAC library is of high quality, as its average insert size is 115 kb with very few (155) blank clones as well as missing wells, and 6X coverage comparable with BAC libraries of other fishes. The average insert size and genome coverage reported in BAC libraries for some important fish are: 80 kb insert size, 9× coverage in pufferfish [21]; 188 kb, 18.8× coverage in salmon [22]; 161 kb, 10.6× in channel catfish [23]; 65 kb, 6× in Nile tilapia [24]; 145 kb, 6× in medaka [25], 121.5 kb, 6.3× in grass carp [26], etc.

The BAC library characterization is a labour-intensive and costly affair. The BAC pooling strategy is a rapid, cost effective method for their characterization and further use in genome assembly, as well as anchoring of the physical maps [27]. A five-dimensional (5-D) BAC clone pooling strategy employing both the GoldenGate assay screening and the assembled BAC contigs was proposed by You et al. [28] for characterization. It is claimed to be a high-throughput, low-cost, rapid and feasible approach to screen BAC libraries and to anchor BAC clones and contigs on the genetic maps. This approach could also reduce the number of PCR procedures. 

The information on BACs is used in many ways, from genome sequencing to gene discovery. There are several databases on BESs in many model species, such as humans, rice, mouse, sea urchin [29], etc. The BAC end sequence information has been reported to be stored as a database, such as BACPAC Genomics (http://bacpacresources.org; accessed on 26 October 2022). The database is helpful to easily identify overlapping clones, select clones for restriction endonuclease fingerprints, identify appropriate clones for FISH mapping, select clone(s) containing genes of interest, etc. We have also developed an offline web server, named “BAC2Genom”, based on BAC end sequences of magur. In channel catfish, approximately 63,366 BESs have been generated for marker development and assessment of syntenic conservation with other fishes [30,31]. Genet et al. [32] generated 176,485 high-quality BES in *Oncorhynchus mykiss* that represented ~4% of the trout genome. In Pacific white shrimp *(Litopenaeus vannamei*), Zhao et al. [33] generated 11,279 BESs (385 bp average sequence length) from 7812 BAC clones, representing 0.18% (4,340,753 bp) of the haploid genome. They successfully annotated BESs and reported genes involved in immunity and sex determination. In addition, their findings also provided an important resource for functional gene studies, map construction and integration, and complete genome assembly of *L. vannamei*. 

The average GC content (38.8%) of *C. magur* BAC clones in this study was closely similar to the GC content (39.2%) of *C. batrachus* genome [34]. The SSRs identified in this study may also be useful for integration of linkage and physical maps, fine QTL mapping and positional cloning of genes for aquaculture traits. The linkage mapping of the SSRs associated with contigs also allows construction of super scaffolds and identification of large conserved genomic segments [31]. There are several studies where BESs were utilised to explore SSRs for the development of markers for various genetic studies. In one study, Bohra et al. [35] partially analyzed 88,860 BESs for SSRs and identified 18,149 SSRs, from which they identified 842 polymorphic SSR markers for their utility in pigeon pea improvement. Xu et al. [30] detected SSRs in 17.5% of BESs. Zhao et al. [33] found di- and hexa-nucleotide repeats as the most common SSR types in the BESs of *L. vannamei*. Genet et al. [32] identified 6848 SSRs in BES analyses, of which 3854 had high-quality flanking sequences for PCR primer design. Lin et al. [36] found 25 SSRs from 17 different BAC clones in tea plants and used them as potential genetic markers. In the present study, 11,082 SSRs could be identified from 36 BAC clones.

It is very cumbersome to find a chromosome-specific marker to ascertain the identity of a chromosome. A BAC clone DNA probe that hybridized on an individual chromosome could be used to identify that particular chromosome. This approach can also be utilised to resolve chromosomal identity in ambiguous morphological conditions. Thus, these BAC clones may be used as a marker to identify the chromosomes. HaiXia et al. [37] constructed an experimental system on *Cyprinus carpio* (var. *specularis*) chromosomes for localizing specific genomic sequences to serve as a powerful tool for carp cytogenetic mapping, genome evolution and comparative genomics research. BAC-FISH has proved to be a backbone to guide the sequencing of tomato chromosome 6 [38]. It was seen that out of eight clones positioned on Scaffold_1, six were hybridized on the same 4^th^ chromosome pair, while two were hybridized on the 16th and 9th sm chromosome pairs. The localization of the same BAC clones on the p arm of one chromosome and q arm of another chromosome indicates the presence of duplicated regions and genes during chromosome events [39] or due to assembly scaffolding issues. HaiXia et al. [37] localized a male-specific marker, CCmf1, from Yellow River carp on at least four chromosomes of mirror carp, suggesting that it might be a repetitive sequence useful for sex-determining gene searching.

BESs and the associated polymorphic markers are a good resource for comparative genome analysis. Dong et al. [12] used BAC-FISH to establish a cytological map to recognise individual chromosomes in *Saccharum spontaneum* and established a relationship with Sorghum. Genet et al. [32] performed comparative genomics using paired BESs of *O. mykiss* and found several regions of conserved synteny across the genome and protein databases of zebrafish, medaka and stickleback. The discovery of specific repeat elements will facilitate analyses of sequence content (e.g., for SNP discovery and transcriptome characterization) as well as future genome sequence assemblies. They opined that comparative genomics through the BES approach can be used for identifying positional candidate genes from QTL mapping studies, aiding in future assembly of a reference genome sequence and elucidating sequence content and complexity in the genome. The genes found to be localized in the BAC clone through mapping of BESs on scaffolds in this study are well-described and have been compared with channel catfish and zebrafish in terms of gene description and pathway. Liu et al. [31] utilized 43,000 BESs for comparative genome analysis in catfish. Using these BES resources, linkage maps and existing physical maps, they identified conserved syntenic regions between catfish and zebrafish genomes. A total of 10,943 catfish BESs (17.3%) had significant BLAST hits in the zebrafish genome, of which 3221 were unique gene hits, providing a platform for comparative mapping based on locations of these genes in catfish and zebrafish. Xu et al. [30] also anchored channel catfish BACs with the zebrafish and *Tetraodon* genome sequences by BLASTN and found 16% and 8.2% significant hits, respectively, where a total of 1074 and 773 significant hits were unique to zebrafish and *Tetraodon* genomes, and of which 417 and 406 were identified as known genes in other species. This provides a major genome resource for comparative genome mapping. Similarly, in the present study, 31 BAC clones of magur containing 86 genes were in synteny with channel catfish and zebrafish. Of these, 26 genes present on 10 magur clones exhibited conserved synteny, while 60 genes distributed on 21 clones had disrupted synteny with channel catfish and zebrafish with locations on different sites.

The approach used in the present study for characterization of BAC clones of *C. magur*, using BAC end sequences mapped to genome scaffold and identifying the size of insert DNA, genes and SSRs and BAC-FISH on the chromosomes, has generated important information. Chromosome-specific magur BACs may also be useful in taxonomic identification of magur hybrids that have been seen recently in catfish aquaculture.

## 4. Material and Methods

### 4.1. Chromosome Preparation

Healthy live specimens (*n* = 20) of magur were collected from a local pond in Lucknow, Uttar Pradesh, India, with the help of fishermen and transported to the lab in live condition. The specimens were identified using taxonomic keys [40]. The metaphase chromosome spreads were prepared in vivo from the anterior kidney cells after colchicine treatment using a standardized protocol [41] for FISH.

### 4.2. BAC Library Construction

The BAC library was constructed under a research project funded by the Department of Biotechnology, Government of India, New Delhi, using high molecular weight (HMW) genomic DNA of magur. The DNA was digested with *HindIII* restriction enzyme and size selected. The selected DNA fragments with an average size of around 150 kb were ligated into pCC1BAC vector (Epicentre Biotechnologies, Madison, WI, USA), which was then transformed in the Phage Restraint DH10β competent cells of *E. coli* (Invitrogen, Burlington, ON, Canada) and propagated. The transformed BAC clones were robotically picked up with a Genetix Qpix 2 Automated Arraying Bacterial Colony Picker (Molecular Devices, Sunnyvale, CA, USA), placed in plates and maintained in Luria Broth (LB) media. 

### 4.3. BAC Clone Culture, DNA Isolation 

We selected clones randomly from the BAC library but preferred that the clone may contain genes to develop physical gene maps of the chromosomes. The selected clones were taken from the preserved BAC library and then cultured in duplicate in 15 mL conical centrifuge tubes containing 8 mL LB medium, 30 µL clone, 8 µL of chloramphenicol antibiotic (12.5 µg/mL conc.) and 3.2 µL of MgSO_4_.7H_2_O (0.4% conc). As the pCC1BAC vector is present in single copy per cell and is inducible to 10–20 copies per cell (https://ecoliwiki.org/colipedia/index.php/pCC1BAC; accessed on 19 May 2020), the culture was performed in duplicate to increase the concentration of insert-DNA. The mixture was then incubated overnight at 37 °C and 231 rpm in shaker incubator for culture. The BAC DNA isolation of selected clones was conducted using the protocol of Kumar et al. [42] to construct a DNA probe and perform FISH for physical localization on chromosomes. The quantity and quality of isolated DNA were checked in NanoDrop 2000 (Thermo Fisher Scientific, Wilmington, DE, USA) and in 0.4% agarose gel. 

### 4.4. BAC End Sequencing

Both ends of BAC clones were sequenced on an ABI 3500 Genetic Analyzer (Thermo Fisher Scientific, USA) using T7 forward (5′-TAATACGACTCACTATAGGG-3′) and pbRP1 reverse (5′-CTCGTATGTTGTGTGGAATTGTGAGC-3′) primers (http://www.epibio.com; accessed on 11 December 2022). The BESs were then mapped on the magur genome scaffolds and analysed with a custom Perl script using Blast tool. The clones, positioned on the same scaffolds, were pooled in single culture for insert-DNA isolation, probe construction and FISH.

### 4.5. Probe Labelling and BAC-FISH

For DNA probe construction, 1 µg BAC isolated insert-DNA from each clone was taken. The DNA was then labelled with green colour fluorescein-12-dUTP (Fermentas, Vilnius, Lithuania) and red colour tetramethyl-rhodamine-5-dUTP (Roche, Basel, Switzerland) fluorophores using the direct labelling technique “nick translation”. Dual colour FISH was performed as per protocol [42] to hybridize probes on the metaphase chromosomes. After hybridization, the chromosomes were counterstained with VectraShield mounting medium (Vector Labs, Burlingame, CA, USA) containing DAPI and antifade for 60 min. Slides were then examined under a fluorescence microscope (Leica, Wetzlar, Germany) for visualization of chromosome and DNA probes with three band filters, i.e., DAPI filter (excitation at 340–380 nm, emission at 461 nm) for chromosome visualization in blue colour, N2.1 filter (excitation at 515–560 nm, emission at 595–605 nm) for rhodamine-labelled probe visualization in red colour and I3 filter (excitation at 450–490 nm) for fluorescein-labelled probe visualization in green colour. Images of both the chromosomes and the probes were acquired and super-imposed on each other using Karyo4000 software (Leica, Germany) to observe the correct position of the probe signals on specific chromosomes. Around 50 metaphase spreads of each clone were analyzed for BAC-FISH hybridization. Karyotypes were then constructed for individual hybridized BAC clones, and the consensus chromosome-containing probe was built. Each BAC-FISH experiment was performed in triplicate.

### 4.6. Functional Annotation of BAC Mapped Genes

The BESs of 36 hybridized clones were aligned on the magur genome assembly (NCBI’s Genome Acc. No. QNUK00000000) using BLASTN tool with e-value 10^−5^. Gene annotation of clones was performed using OmicsBox tool against the UniProt and NR databases, followed by InterPro mapping using InterProScan database integrated with OmicsBox. The simple sequence repeat (SSR) information of the clones was mined using WGSSAT tool [43] integrated with MISA [44] in its background. 

### 4.7. Comparative Genomics

The synteny analysis of each annotated clone gene was performed manually by searching the Ensembl and NCBI databases using zebrafish and channel catfish genomes as query, followed by retrieval of the location of each gene with chromosome information. A Circos plot [45] was used for synteny visualization, taking only the genes’ regions present on the same magur scaffolds but on different chromosome locations in channel catfish and zebrafish. If the genes were present on the same magur scaffold and on zebrafish and channel catfish chromosomes, then only one gene from the group was taken to construct synteny to make the visualization clear. Panther database [46] was used for gene ontology enrichment and to derive relationships between the genes. KEGG Mapper tool [47] was applied for pathway mapping.

The sequences of common genes present on channel catfish and zebrafish (retrieved from Ensembl genome database) and *C. magur* scaffolds were used for phylogenetic analyses. The neighbour-joining (NJ) method based on p-distances was used (as similar DNA sites among the three species were absent) for phylogenetic analysis using MEGA software [48]. The interactive tree of life (iTOL) v5.5 tool [49] was used for the presentation of the phylogenetic topology of DNA sequences for estimating evolutionary lineages.

## 5. Conclusions

The construction of BAC libraries and their characterization through BESs provided insights into the *C. magur* genome. In this study, we identified chromosome-specific BACs through FISH. It is an important step in the progress towards a modest and reproducible process for chromosome identification using cytogenetic markers with BAC-FISH. This is the first establishment of a BAC-FISH based magur physical map. The information generated in this study will aid in the improvement of genome assembly for magur and, thereby, provide a platform for chromosome level assembly. The gene of interest on the chromosome may be used to validate as well as enrich the genome assembly and help in reconstruction of the order of the genes and genetic markers present in magur and other related genomes. The generated information may be used in a wide array of applications, such as the conservation genetics of the endangered catfish magur and other clariid species through molecular cytogenetic research, including gene localization, karyotype analysis, positional cloning, etc. 

## Figures and Tables

**Figure 1 ijms-23-15958-f001:**
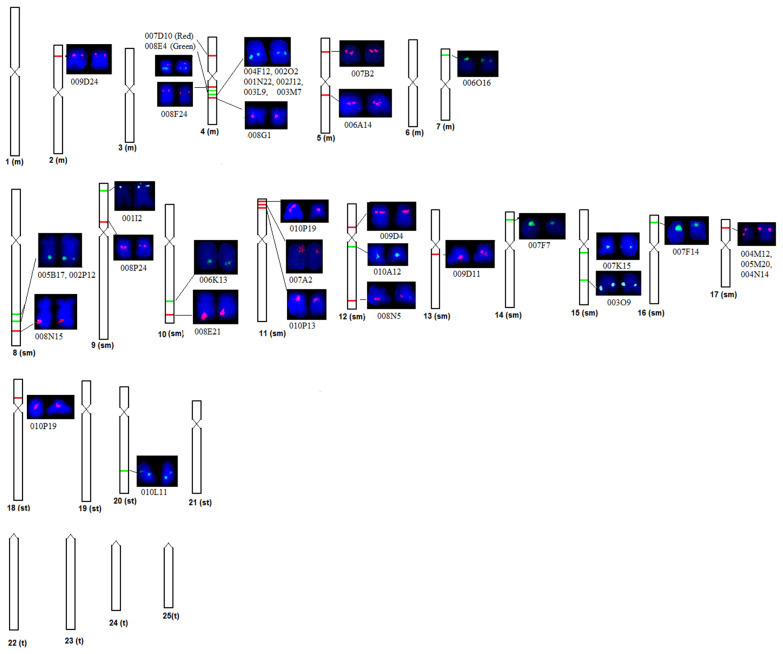
BAC-FISH of 36 clones on metaphase chromosomes of *Clarias magur.* Signal with red colour was DNA probe labelled with Tetramethyl rhodamine-5 dUTP and green colour labelled with Flourescien-12 dUTP. Signals with clone ID represent the physical location of BAC in the corresponding region of magur chromosomes.

**Figure 2 ijms-23-15958-f002:**
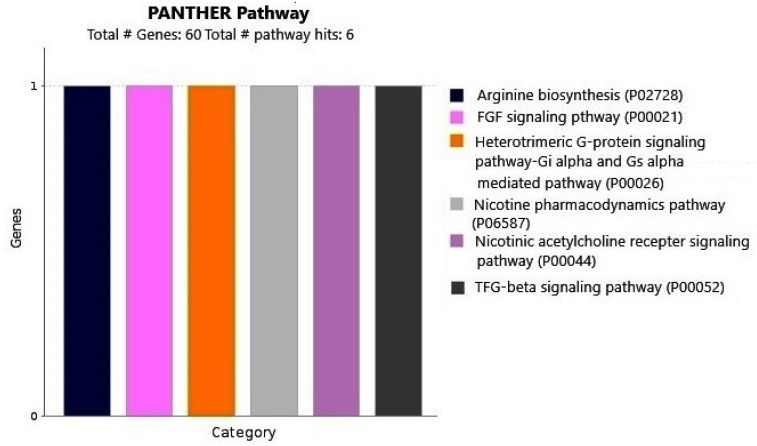
Panther database analysis showing pathway analysis of genes present on 36 BAC clones.

**Figure 3 ijms-23-15958-f003:**
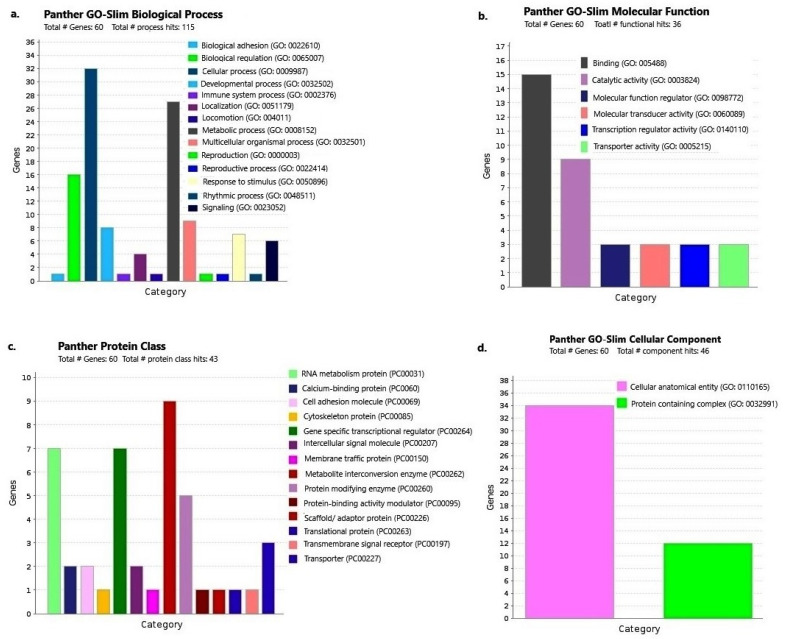
Panther database analysis showing GO terms associated with: (**a**) biological process, (**b**) cellular component, (**c**) molecular function and, (**d**) protein class of 36 studied clones.

**Figure 4 ijms-23-15958-f004:**
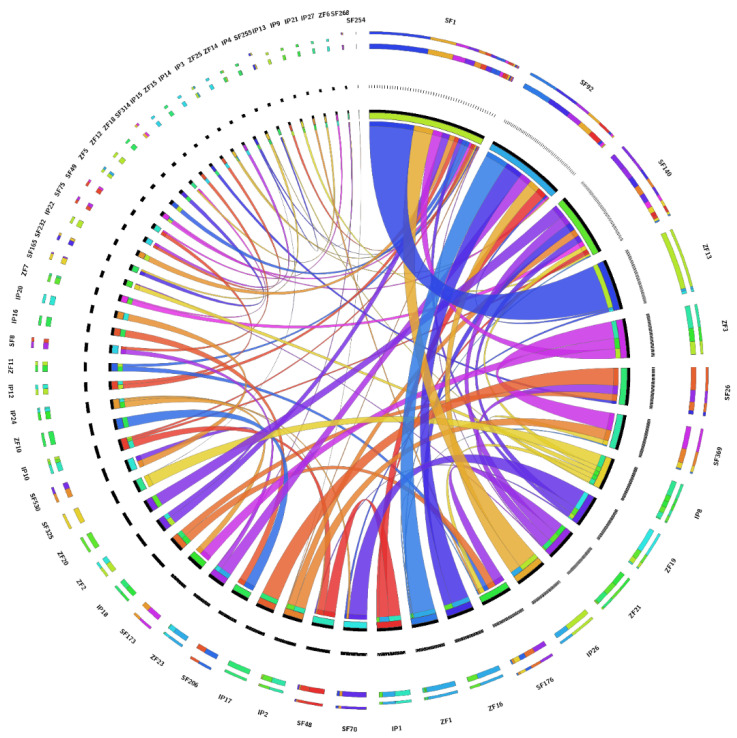
Synteny visualization of 45 genes present on 21 genome scaffolds of magur with channel catfish and zebrafish.

**Figure 5 ijms-23-15958-f005:**
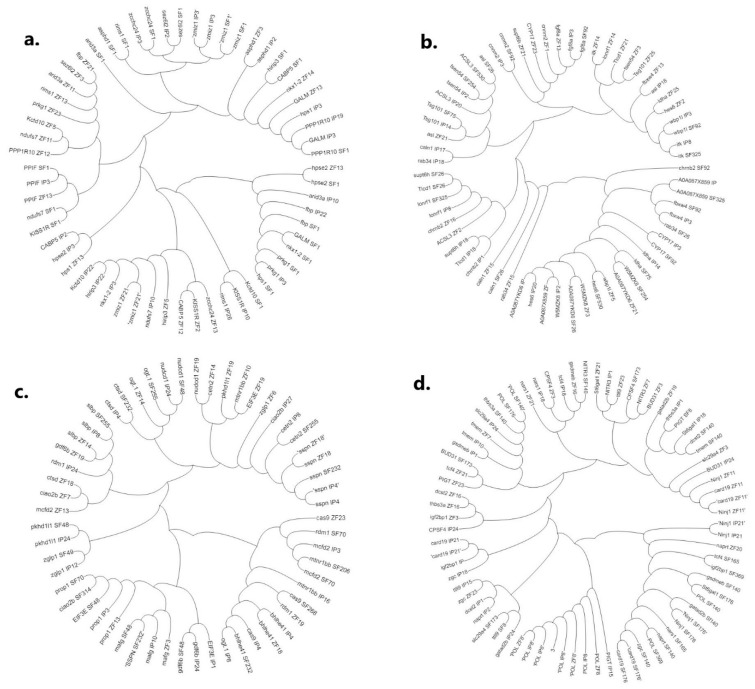
Phylogenetic trees constructed by neighbour-joining method based on *p*-distance of common genes present on chromosomes of channel catfish and zebrafish with *C. magur* scaffold: (**a**) SF_1; (**b**) SF_26, SF_75, SF_92; SF_254, SF_325, SF_530; (**c**) SF_48, SF_49, SF_70, SF_206, SF_232, SF_255, SF_268, SF_314; and (**d**) SF_8; SF_140, SF_165, SF_173, SF_176, SF_369.

## Data Availability

Not applicable.

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
