# Peer review of "BAC-FISH Based Physical Map of Endangered Catfish Clarias magur for Chromosome Cataloguing and Gene Isolation through Positional Cloning"

_ijms, 2022, doi:10.3390/ijms232415958_

Round 1

Reviewer 1 Report

The paper brings an important contribution to the knowledge of Clarias magur genom.  Mapping using BAC-FISH probes allows, in a straightforward and easy way, the final assembly and mapping of the genes in their chromosomes. The authors produced a karyological physical map and identified a wide range of functional or metabolic genes, some of them very important in survival and reproduction of the species. Moreover, by using different bioinformatics tools the authors managed to provide a comprehensive interspecific comparison between the genes’ regions on magur chromosome scaffolds and those of channel catfish and zebrafish. However, we suggest to the authors to perform an extensive revision of the text especially the introduction and Discussion sections in order to remove some redundant information and increase the text clarity. In Discussions you need to better highlight your contribution while you compare your results with other similar studies. Also, please proofread the English.

Reviewer 2 Report

In this manuscript by V. S. Baisvar and colleagues, the authors present a study of BAC clones selected from BAC library of Clarias magur. The clones were bioinformatically mapped on the magur genome scaffolds, used for BAC-FISH. analyzed for the presence of genes and simple sequence repeats. Sequences of these genes were aligned with zebrafish and channel catfish genomes in a comparative study. I found this manuscript interesting and suitable for publication in International Journal of Molecular Sciences. However, before accepting it for publication, several issues should be considered by the authors.

Results

Figure 1. - line 110: blue colour labelled                  change to green colour labelled

Supplementary Table 1: columns G and H are not consistent when the BAC clone is located on two scaffolds (lines 1, 2, 5, 12, 13, 17, 21). Correct column G in lines 1, 2, 5, 12, 13, 17, 21.

Discussion

As written in the results, 8 clones positioned on Scaffold_1 hybridized on 3 different chromosomes („ Out of 8 clones positioned on Scaffold_1, six were hybridized on 4th chromosome pair, while one clone was hybridized on 16th chromosome pair and one clone on 9th sm chromosome pair“). It would be useful to discuss this also in the discussion paragraph.
